# Super-Pixel Guided Low-Light Images Enhancement with Features Restoration

**DOI:** 10.3390/s22103667

**Published:** 2022-05-11

**Authors:** Xiaoming Liu, Yan Yang, Yuanhong Zhong, Dong Xiong, Zhiyong Huang

**Affiliations:** School of Microelectronics and Communications Engineering, Chongqing University, Chongqing 400044, China; lxm@ccee.cqu.edu.cn (X.L.); 20191202021t@cqu.edu.cn (Y.Y.); xiongd@cqu.edu.cn (D.X.); zyhuang@cqu.edu.cn (Z.H.)

**Keywords:** low-light, Image enhancement, attentive neural processes, super-pixel segmentation

## Abstract

Dealing with low-light images is a challenging problem in the image processing field. A mature low-light enhancement technology will not only be conductive to human visual perception but also lay a solid foundation for the subsequent high-level tasks, such as target detection and image classification. In order to balance the visual effect of the image and the contribution of the subsequent task, this paper proposes utilizing shallow Convolutional Neural Networks (CNNs) as the priori image processing to restore the necessary image feature information, which is followed by super-pixel image segmentation to obtain image regions with similar colors and brightness and, finally, the Attentive Neural Processes (ANPs) network to find its local enhancement function on each super-pixel to further restore features and details. Through extensive experiments on the synthesized low-light image and the real low-light image, the experimental results of our algorithm reach 23.402, 0.920, and 2.2490 for Peak Signal to Noise Ratio (PSNR), Structural Similarity (SSIM), and Natural Image Quality Evaluator (NIQE), respectively. As demonstrated by the experiments on image Scale-Invariant Feature Transform (SIFT) feature detection and subsequent target detection, the results of our approach achieve excellent results in visual effect and image features.

## 1. Introduction

Image enhancement technology has gradually pervaded all aspects of human life and social production [1], with the enhancement of the low-light images being an important branch of image enhancement. In the real scene, due to the lack of ambient light, anomalous exposure, and other factors, when we took dark images, as shown in Figure 1a–c, the low light environment affects our observation, and it greatly affects computer vision tasks such as target recognition and landscape analysis [2]. Relatively, we select some of the bright images of the Pascal VOC dataset (https://pjreddie.com/projects/pascal-voc-dataset-mirror/ (accessed on 1 March 2020)) to display in Figure 1d–f, and we can see that the bright images have a uniform distribution of light and have distinct features with rich detailed information. Compared to dark images, bright images usually provide humans a pleasant visual enjoyment and allow computers to better simulate human perception and observation to obtain more accurate information about the corresponding scenes. Therefore, we are looking for ways to improve the “quality” of low-light images for specific visual tasks. There are two quality requirements: visual effect and image features. The visual effect of an image influences a person’s most intuitive evaluation of the image: for example, an image with full color and moderate brightness will often bring people a pleasant experience. Image features describe the characteristics of the image or its surrounding area [3], which can influence how an image is interpreted and recognized for a subsequent special task. Although there are many studies on low-light image enhancement [4,5,6], their methods often fail to achieve a good balance between visual effects and image features, ignoring feature information to make the image aesthetically pleasing.

Figure 1a,b show that real-world low-light images near the light source are sharper and contain more image information than those far away and that the number of features contained in each image location varies greatly in both in bright and low-light images. As a result, if we use the same enhancement strategies for every region of an image, we will frequently end up with an overexposed or still dark image that ignores local information. Most of the existing methods focus on the global strategies, Liang [7] takes this into account and chooses the Gaussian Process (GP) to enhance low-light images. However, because the method is based on uniform patches, the results are unsatisfactory. Inspired by [7], this paper utilizes super-pixel image segmentation and ANP as the key enhancement step to manipulate low-light enhancement from global to local. The super-pixel image segmentation can divide images based on their luminance and color similarity, and the regions of the segmentation can be improved with ANP to achieve a more natural visual effect and restore the local features. In comparison to GP, ANP incorporates the attentional mechanism, which improves the effectiveness of its observations of relationships between image regions.

Our approach involves the following steps. To begin, we train a CNN network using a synthetic low-light image dataset to produce a preliminary enhanced image as an a priori image processing. This process attempts to restore as much of the original low-light image feature information as possible. After that, we use super-pixel image segmentation to create a series of image regions made up of pixels with adjacent positions and similar features such as color, brightness, texture, etc. Following that, ANP trains each low-light super-pixel to improve its brightness, emphasize and recover its features, and then fuse them to create the final enhanced image. Each ANP training includes the results of the previous training to eliminate the unnatural junction part of each image region during the stitching and fusion process. The experimental results show that in comparison to other advanced techniques, our framework improves the visual brightness of the image while also restoring the image’s feature information for subsequent tasks.

The main contributions of this paper are as follows:We use CNN as the initial enhancement step, which can restore features from the original image, allowing us to retain more details in our enhancement results for subsequent feature matching, target detection, and other image processing operations.We propose a new local image enhancement method that utilizes super-pixel segmentation to obtain image regions and then enhances local information with ANP networks.Through extensive experiments on the synthesized low-light image and the real low-light image, we verify that our approach achieves excellent results in visual effect and image features.

This paper is organized as follows. Section 2 details the classification of low-light image enhancement algorithms and the advantages and disadvantages of common algorithms. Section 3 presents our experimental framework by modules in detail. Section 4 presents the results of the study. Section 5 is the conclusion of this paper.

## 2. Related Works

Computer vision tasks are a rapidly developing branch of artificial intelligence that is widely used in various engineering fields [8,9]. Chen [10] established a four-camera vision system to obtain the visual information of targets and designed a point cloud correction algorithm by filtering and splicing operations. Tang [11] proposed a mathematical model for reconstructing the 3D deformation surface using the four-ocular visual coordinates and point cloud matching.

Low-light image enhancement techniques are an important aspect of computer vision and are critical for subsequent higher-order image tasks. Low-light image enhancement is a technique for improving the brightness, contrast, color, and detail of low-quality images taken in a low-light environment. At present, the various works can be categorized into four categories [2].

The first category includes methods based on Histogram Equalization (HE) [12,13,14,15]. The basic concept behind HE is to alter the grayscale of the pixels in the original image. It widens the grayscale levels in the image with more pixels and narrows the grayscale levels in the image with fewer pixels, transforming the corresponding histogram into a uniformly distributed form. The purpose is to enhance the overall contrast of the image to improve its clarity. Moreover, the image reaches the greatest entropy and contains the greatest amount of information. The Contrast Limited Adaptive Histogram Equalization (CLAHE) [12] method proposed by Pisano et al. is a type of local processing of images that prevents over-enhancement by limiting the stretching of similar grayscale levels in local image patches.

The second category includes methods based on the Retinex model [16,17,18,19]. The basic theory behind the Retinex model is that the color of an object is determined by its reflective ability to long-wave (red), medium-wave (green), and short-wave (blue) light. Its color is not determined by the absolute value of the intensity of the reflected light and is unaffected by the non-uniformity of illumination. It means Retinex is based on color consistency. Unlike traditional linear and non-linear methods that can only enhance certain types of image features, Retinex can achieve a balance between dynamic range compression, edge enhancement, and color constancy, allowing for adaptive enhancement of a variety of different types of images. Li et al. [16] combine the noise map and the Retinex model to achieve low-light enhancement by proposing a new optimization function.

The third category includes methods based on the defogging model [20,21,22,23]. Kaiming He proposed the a priori theory of dark channels in images in 2011 [20], which has been widely used in the field of image enhancement. The algorithm’s main idea is that inverting a color image taken in a dark environment has a similar visual effect to a daytime image taken in a foggy environment. Therefore, the inverted low-light image can be processed by using the defogging algorithm based on a dark channel prior, which can then be inverted back to obtain an enhanced low-light image.

The fourth category includes methods based on neural networks, which have made significant advances in image processing in recent years [24,25,26,27]. Deep learning is the creation of a network model that mimics the human brain’s information processing mechanism. It utilizes an efficient learning strategy to gradually restore features to accommodate complex non-linear functions. CNN is been used as the basis of deep learning frameworks in many research works [24,26]. Furthermore, deep learning approaches have been proposed based on the Retinex theory as well [25].

The methods listed above have worked well in the low-light field in different ways. However, there are still some issues that need to be solved. The HE model has the disadvantage of causing color shift and a loss of detail in the image due to grayscale merging. In some scenes, the Retinex models can compromise the image’s robustness and cause severe color distortion. The defogging model lacks physical interpretation, while the brightness and contrast of the resulting graphs still need to be improved. Deep learning-based models for low-light image enhancement have performance advantages, but they require large datasets, and the increased complexity of the network structure leads to a sharp increase in the time complexity of the corresponding algorithms. Furthermore, the majority of methods focus solely on the visual enjoyment provided by the final result, ignoring the critical information such as image features required for subsequent image processing. However, features are crucial for images to enhance the robustness of image matching and improve the efficiency of subsequent image target matching.

Our paper seeks to find a balance between image visualization and image feature restoration and improve the ability of our approach to combine fast operation and good enhancement quality. Therefore, our method utilizes CNN as an image prior; then, it selects super-pixel segmentation and ANP as image local enhancement steps, which differs from traditional neural networks that look for a set of function relations and then invokes different mappings according to different requirements to obtain the final result rather than looking for a global function to represent the enhancement relations. This makes our experimental results not only visually appealing but also restores the original feature information of the image and enriches the image details.

## 3. Proposed Method

In this section, we present give the overall framework of our algorithm, which contains two stages. We then describe in detail the component modules of the two stages.

### 3.1. The Overall Framework of Our Method

Our method is divided into two stages. The first stage is to obtain the image’s initial global enhancement result. We use a shallow CNN to improve the brightness of the original image while restoring primary features. The second stage is the local enhancement of the image. We focus on the use of super-pixel segmentation and the ANP network structure in our approach to achieve the corresponding enhancement of different image regions. The local enhancement step can help restore low-light image features and improve visual effects.

First, the RGB color space is converted to YUV color space for the low-light image. We then use a CNN network to obtain the preliminary enhancement results for the luminance component (Y) of the low-light image while keeping the chromaticity (U) and saturation (V) components unchanged. Then, we perform super-pixel segmentation on the initial enhanced image obtained by CNN and copy the segmentation result directly to the corresponding low-light image to have consistent regions of segmentation between image pairs. Finally, each low-light image region is gradually trained with ANP to achieve natural image blend boundaries, and then, the output image is converted to RGB space to obtain the final enhanced image. The flowchart of our algorithm is shown in Figure 2.

### 3.2. Initial Enhancement

Since bright images contain rich feature information, as shown in Figure 1d–f, we utilize bright images to generate low-light images as a training set, which is generated with a fixed formula. Therefore, we designed a shallow neural network to recover the bright images and consequently restore the features. Figure 3 shows the CNN network structure for our approach.

There are four convolutional layers in the CNN network, with the kernel sizes 9×9, 1×1, 5×5 and 5×5, and output feature maps 64, 32, 16, and 1, respectively. We choose to minimize the Mean Square Error (MSE) loss function to update the weights to establish pixel-to-pixel correspondence and complete the mapping between images. It is expressed as Equation (Equation 1):(1)$$MSE=\frac{\sum_{i=1}^{n}\left(y_{i}-{y_{i}}^{p}\right){}^{2}}{n}\;$$

*N* in the equation represents the number of pixels in the image, and $y_{i}$ and ${y_{i}}^{p}$ represent the intensity of pixel *i* in the training image and the corresponding target image, respectively.

The Linear Rectification (ReLu) and Leaky ReLu (LReLu) functions [28] were chosen because of their highly non-linear characteristics, which allows them to mitigate the gradient vanishing phenomenon. When these two functions are combined, they can improve the ability to retain the original image’s detailed information and facilitate feature restoration using high-speed training.

In this paper, we use a CNN network to determine the Y component of a bright image based on its corresponding low-light image, which requires a large amount of supervised data. However, the current datasets of low-light images [25] are usually images taken under low-light conditions or low-light images obtained by changing the camera exposure, which is not conducive to the training of the CNN network. Furthermore, because the bright images, as mentioned in Section 1, contain rich feature information, they can be used as ground truth to generate low-light images to create a paired dataset. It is then possible to design a CNN network to recover the ground truth image and thereby restore the features indirectly. We adopt the method of adjusting normal images with Equation (Equation 2) based on gamma correction to generate low-light images with different darkness levels.
(2)$$V_{out}=V_{in}^{\gamma }\;$$

$V_{out}$ represents the output image, while $V_{in}$ represents the input image. The whole process is pixel-wise. γ is utilized to adjust image illumination. We choose γ=[2,4,8,16] to perform the synthesis of low-light images, and the effect is shown in Figure 4.

### 3.3. Local Enhancement

#### 3.3.1. The Super-Pixel Segmentation

Super-pixel [29] refers to the irregular blocks of pixels with similar texture, color, brightness, and other features, which make up a certain visual meaning. Super-pixel segmentation is the process of grouping pixels together based on their similar features and replacing a large number of pixels with a small number of super-pixels to express an image’s features, greatly reducing image post-processing complexity. Super-pixel segmentation of the image, in our approach, makes subsequent image local enhancement operations easier, resulting in a more natural visual result with more details.

Our paper uses Simple Linear Iterative Clustering (SLIC) [30], a simple linear iterative clustering method, to segment the preliminary enhanced images obtained by CNN. Then, based on the segmentation result, we acquire the same regions on the low-light image to form region pairs as ANP training data. CNN networks can improve low-light images by restoring features in the dark areas, resulting in more accurate super-pixel segmentation results. However, there is no corresponding bright image for real low-light images, resulting in inaccurate segmentation results. So, the initial enhancement of the CNN network for low-light images is necessary for the super-pixel segmentation step in our method.

Some “isolated” pixels that do not belong to the same connectivity component as their clustering center may remain after the clustering process is completed. Therefore, in order to make the color brightness of the regions of images obtained by segmentation more similar, the label of the nearest cluster center can be assigned to these pixels using a connectivity component algorithm [31].

#### 3.3.2. ANP for Local Enhancement

After the super-pixel segmentation, we obtain the image region pairs, utilize the ANP network to learn the function mapping distribution of each regional pair from input to output, and use the distribution to make predictions for a given input when testing.

Gaussian processes are very powerful approximators. DeepMind has presented three studies that combine Gaussian processes with neural networks, the models that achieve the efficiency of neural network training with the flexibility of Gaussian processes in inference, namely Neural Processes (NPs) [32], Conditional Neural Processes (CNPs) [33], and Attentive Neural Processes (ANPs) [34]. ANP incorporates an attentional mechanism, which result in a better fit than the previous two. ANP can reason about multiple functions for the same data and can capture output synergies for a given input. The overall structure of the ANP network in this paper is shown in Figure 5.

During the training phase, we first get a subset of contexts *C* with randomly selecting *N* pixel pairs at the same location from the initial enhancement image obtained by CNN and the input low-light image. Then, $N_{c}$ pixel pairs are randomly selected from *C* as targets *T*. Through three paths, they obtain the data expression r*, the normally distributed expression *Z*, and the distribution F(T) of *T*. During the testing phase, we assume that the target is set to $T^{\prime }$, which is a sequence of uniformly distributed intensity values between the region’s lowest and highest pixels and is determined by the input image region. When combining this sequence with the trained parameters, we can obtain a distribution of pixel intensities in the region. Finally, we can apply the distribution to the original low-light image region to obtain the final enhancement results.

In the deterministic path, the encoder is shared between all context pairs and consists of a Multilayer Perceptron (MLP) and an attention module. The experimental task finds the correspondence between two-dimensional image intensities, which is essentially a one-dimensional regression task. Therefore, three layers of MLP with ReLu nonlinearities are sufficient to simulate the interaction between each context pair, with the implicit layer’s number of neurons set to 128. The network structure includes an attentional mechanism that allows it to focus on a subset of its inputs (or features) and thus more efficiently selects image information for enhancement. We choose Multihead attention [35] in our paper. It is a parametrized extension that linearly transforms the keys, values, and queries for each head before applying dot-product attention to produce headspecific values. These values are concatenated and linearly transformed to produce the final values as shown in Equation (Equation 3):(3)$$\begin{array}{c} MultiHead\left(Q,K,V\right)=concat\left(head_{1},...,head_{H}\right)W\\ head_{h}=DotProduct\left(Q{W_{h}}^{Q},K{W_{h}}^{K},V{W_{h}}^{V}\right), \end{array}\;$$
where *Q* represents the numerical matrix for a given low light intensity *T*, *K* represents the data matrix extracted during training, and *V* is the data representation of training data obtained through MLP. *H* represents the number of subspaces divided on the model, which is chosen as 8 in this paper. *W* is a weight matrix to indicate the importance of each part of the information.

In the latent path, the encoder in its path is also shared by all context pairs, which has the same hidden layer size as the defined path. Then, using two fully connected layers, we calculate the mean and standard deviation of the data representation to obtain its distribution expression *Z*, which results correlations in the marginal distribution of the target predictions.

In the decoding path, r*, *T*, and *Z* are passed through the decoder to predict the maximum posterior probability value, and the parameters of the whole network are learned the same as in article [34] by maximizing the Evidence Lower Bound (ELBO) [36] as shown in Equation (Equation 4):(4)$$\log p\left(\left.y_{T}\right|x_{T},x_{c},y_{c}\right)\geq E_{q(\left.z\right|s_{T})}\left[\log p\left(\left.y_{T}\right|x_{T},r_{c},z\right)\right]-D_{KL}\left(q\left(\left.z\right|s_{T}\right)∥q(z\left|s_{c}\right.)\right)\;$$
for a random subset of contexts *C* and targets *T* via the reparameterization trick.

When the ANP has enhanced the low-light super-pixel region, the result is combined with the next super-pixel region and sent to the ANP network for progressive enhancement. Then, we repeat this step until the low-light image is completely enhanced. The step-by-step enhancement process causes ANP to pay more attention to the local regions, resulting in more natural stitching and fusion between local regions of the image and also stronger features of each super-pixel region.

## 4. Experiments and Results Analysis

In this section, the performance of our proposed methodology is compared with several state-of-art methods, including: LIME [37], Gaussian Process (GP) [7], CLAHE [38], RetinexNet [25], BIMEF [39], OCTM [40], DeHaze [21], KinD+ [41] and CNN in stage 1. For supervised images, we used PSNR and SSIM [42] as the quality evaluation; as for feature restoration, we use SIFT feature matching to validate our method. The synthetic low-light images are obtained by darkening the brightness of the bright images according to the method described in Section 3.2. For real low-light images, we select the ExDark dataset [43] for qualitative assessment. The Exclusively Dark dataset contains 7363 low-light images with 12 object classes from very low-light environments. We evaluate our method from subjective effects and objective indicators with the NIQE [44] metric and numbers of sift features. In addition, to explore the superiority of our method in feature restoration, we perform object detection on the enhanced results. To establish a fair comparison with the other methods, the code of [7] is reimplemented based on the details given by the paper, whereas the codes of LIME (https://github.com/aeinrw/LIME (accessed on 1 May 2021)), CLAHE (https://github.com/lxcnju/histogramequalization (accessed on 1 May 2021)), RetinexNet (https://github.com/daooshee/BMVC2018website (accessed on 1 May 2021)), BIMEF (https://github.com/baidut/BIMEF (accessed on 1 May 2021)), OCTM (https://github.com/Eason-Sun/Enhancement-of-Low-Lighting-Color-Images (accessed on 1 May 2021)), DeHaze (https://github.com/evmavrop/Hyperion (accessed on 1 May 2021)) and KinD+ (https://github.com/zhangyhuaee/KinD (accessed on 1 May 2022)) are obtained from the open source website.

### 4.1. Implementation Details

For the CNN training data, we selected 300 normal high-resolution images from the Pascal VOC dataset with correction to obtain 1600 images (including the original images) with four different darkness levels as training data. The model was trained using the original images that are resized to 255×255 pixels and normalized to the range of [0, 1]. We chose to train the CNN to restore the image’s contextual information and features, ensuring that it captures valid object features, while ANP handles the subsequent local intensity processing. During each step of the ANP network training process, we randomly select 200 pairs of pixels at a time for a total of 2000 training sessions to obtain the distribution. The experiment uses the tensorflow1.6 and, python3.6 environments, and the experimental platform includes an Intel (R) Core (TM) i5-9300H processor, 8 GB of-G memory, and an NVIDIA GTX 1650 GPU.

### 4.2. Supervised Images Qualitative Evaluation

Firstly, we qualitatively compare the results of the methods in this paper to a number of currently available methods on supervised images. Figure 6 shows an example of a low-light image and the results produced by each method.

It is clear that our method, along with GP and CNN, has the most natural and noticeable brightness. In addition, for the part with more details and more complex lines in the image (blue boxes in Figure 6), RetinexNet, LIME and Dehaze’s result has a certain degree of feature loss, and the colors are too close to each other, which makes it difficult to distinguish the details of the image. CLAHE has some color distortion, OCTM enhances the brighter part of the original image, which produces unsatisfactory results. KinD+ and BIMEF’s overall enhancement effect is not as significant as our method. To make the comparison of results more convincing, more examples of results are shown in Figure 7.

Figure 8 is the partially image patches obtained by enlarging the red box of Figure 7, and it can be seen that CNN highlights the feature boundaries of the original image to provide key information. However, its overemphasis on features leads to unnatural color transitions in the image. Subsequent ANP enhancement can preserve the details of the original image in a more natural way. In comparison to GP, ANP’s attention mechanism makes it more predictive of function distribution and reduces the appearance of noise. The results can also be reflected in subsequent objective metrics.

In order to objectively assess the image quality, we use quantitative assessments based on three evaluation metrics: the PSNR, SSIM, and local feature matching.

PSNR (Peak Signal to Noise Ratio) is an objective standard for evaluating images. To measure the quality of the processed image, we usually refer to the PSNR value to see if a method is satisfactory. Its mathematical formula is as shown in Equation (Equation 5):(5)$$PSNR=10\times \log_{10}\left(\frac{{\left(2^{n}-1\right)}^{2}}{MSE}\right),$$
where MSE is the mean square error between the original image and the resulting image.

SSIM (Structural SIMilarity) is a measure of the similarity between two images, one of which is an uncompressed, distortion-free image, and the other is a distorted image. Given two images *x* and *y*, the SSIM of the two images can be determined as shown in Equation (Equation 6):(6)$$SSIM\left(x,y\right)=\frac{\left(2\mu_{x}\mu_{y}+c_{1}\right)\left(2\sigma_{xy}+c_{2}\right)}{\left({\mu_{x}}^{2}+{\mu_{y}}^{2}+c_{1}\right)\left({\sigma_{x}}^{2}+{\sigma_{y}}^{2}+c_{2}\right)},$$
where μ denotes the mean and σ denotes the variance of the images. $c_{1}$ and $c_{2}$ stand for two constants, avoiding division by zero. When two images are identical, the value of SSIM is equal to 1.

Table 1 shows the results of the average PSNR for the RGB channel and the SSIM for the gray channel for all test images. All values were obtained by selecting 50 low-light synthetic images at random and enhancing them in the same running environment. To ensure fair results for comparison, we used the same parameters on the evaluation metrics. In particular, as part of the ablation experiment, we add the network with the super-pixel segmentation step removed to verify the functionality of our individual modulesas.

From Table 1, it can be seen that our method has the highest PSNR and SSIM. Our method is more natural and produces a better visual effect, which also means that ANP prioritizes the extraction and restoration of details and features over other network structures without sacrificing the enhancement effect. In addition, according to the ablation experiments, we can find that super-pixel segmentation plays an important role in the whole algorithm. It enables each image processing to focus on a specific region, reducing the amount of data per processing and improving the effectiveness of feature recovery.

For qualitative evaluation of image features, we use SIFT feature matching [7], which is scale-invariant feature transformation, to detect local features of the image, and extract the position, scale, and rotation invariance of these points. These key points are some very prominent points that do not change due to lighting and noise factors, such as corner points, edge points, bright spots in dark areas, and dark spots in bright areas, so they are independent of the image size and rotation, and they have a high tolerance for light, noise, and perspective changes. Figure 9 shows some examples of features detected from the enhancement results and matched to the features detected in the original bright image. We use Euclidean distance to describe the similarity between features and equally set the threshold to 0.4. We can also see from the ablation experiments that super-pixel segmentation is important in the overall algorithm. It allows each image processing to focus on a specific region, reducing the amount of data processed per processing and improving feature recovery effectiveness.

The image’s feature points are represented by the colored circles in the figure, which represent the image’s color, texture, shape, and spatial relationships. Although visually appealing, the results obtained by LIME and RetinexNet suffer from color distortion, so the feature point matching will differ significantly from the original image. It can be seen that our method has the most feature points that match the original image, implying that it achieves the feature restoration effect. Furthermore, when compared to the CNN results in Stage 1, the addition of ANP improves the naturalness of the enhancement results and restores more features.

### 4.3. Real Low-Light Images Qualitative Evaluation

Realistic low-light images usually do not have a corresponding image to compare. Therefore, we will provide three unsupervised evaluation criteria in this section.

First of all, for the subjective evaluation, we selected 30 people for the optimal choice of the results of different methods, and the statistical results are shown in Figure 10. The low-light images for the survey were chosen at random from the ExDark dataset.

As can be seen in Figure 10, the results obtained by our method are most widely accepted and more in line with the public’s visual perception. According to multiple reflections, LIME is overexposed, BIMEF and Dehaze enhancements are relatively dim, KinD+ makes the image contrast too high and blurs the image details, and the other methods are distorted. To get a better sense of each method’s visual effect and to match the previous subjective evaluation, we show the results of three different real low-light images a,b,c obtained through different enhancement algorithms in Figure 11.

Many methods are available to perform unsupervised quality assessment of images, such as the Deep Neural Network-Based Super-Resolution Image Quality Assessor (DeepSRQ) [45,46]. In this paper, we choose NIQE [44] as the objective evaluation metric in this section, which does not require the original image and uses the Multivariate Gaussian (MYG) model to normalize the modeled image pixels. NIQE has a higher level of predictability, monotonicity, and consistency. It is more similar to the human visual system and can evaluate image quality in real time. The NIQE results are assessed as shown in Table 2. The smaller the NIQE, the better the image quality.

The values in Table 2 are the results of randomly selecting 50 real low-light images from the ExDark dataset for enhancement and then averaging the NIQE for each effect. When compared to other methods, we can see that our method has the lowest NIQE value, indicating that our algorithm properly adjusts the light and significantly eliminates degradation to achieve excellent results.

Finally, we use the number of features approach to evaluate the ability of feature restoration in enhanced images, which is an important metric for evaluating image quality. In this experiment, we utilize SIFT to implement keypoints detection, and the number of features acquired is represented in Figure 12. The edge response points are removed in all experiments to avoid the effects of noise on feature recovery. As can be seen, our method recovers the most features from low-light images, implying that our final result has the most edge and detail information.

### 4.4. Object Detection Test

In this section, we will further investigate the potential of low-light image enhancement in advanced computer vision tasks such as target detection. For this purpose, we utilize Yolov3 [47] to test on real low-light images as well as their enhancement results obtained by LIME, RetinexNet, the first stage of CNN, GP, KinD+, and our method. We do not adjust or retrain any parameters of the Yolov3 model to obtain the most realistic and intuitive comparison.

From Figure 13, we can see that our method has the best detection results. For instance, in the left example, our method allows the two most prominent objects in the image, the car and building, while the others are overly concentrated in the red area, leading to the misjudgment of fire or misidentification of noise obtained during the low-light enhancement as fireworks or other objects. In the right example, our method accurately detects the presence of buses, not cars, boats, or other potentially confusing objects, and it is the only method capable of detecting the presence of streetlights. We can see that the addition of the local enhancement method in stage 2 makes the image information extracted by CNN more prominent, and the image detection results are more in line with the ground truth.

The experiment demonstrates the viability of low-light image enhancement in practical application, demonstrating and showing that our proposed enhancement technology can be used as a pre-processing method for tasks such as target detection, image classification, etc.

## 5. Discussion

In this paper, we try to make the low-light image enhancement results not only be a pleasant visual experience for the observer but also restore as much feature and image detail as possible for subsequent image vision tasks. We utilize CNN trained by synthetic data as the image prior to obtaining the initial enhanced image. Then, at the image local enhancement step, we use super-pixel segmentation and ANP to achieve a more natural view and richer image detail. Furthermore, the experiments on the synthesized low-light image and the real low-light image show that our method achieves the state-of-the-art visual effects and capability of feature restoration. Our method varies from typical low-light enhancement methods in that it places a greater emphasis on picture feature recovery, which is critical for subsequent computer image processing. Although the color of our result is not sufficiently sharp, we have a more natural visual effect. Nonetheless, there are several limitations to our approach. Our network structure’s overall computational load is currently quite high. The time it takes us to improve an image is consistently 2 min, which is substantially longer than other algorithms’ second or even millisecond accuracy. We will continue to innovate ANP’s structure and combine the complete network into an end-to-end architecture in future work. In addition, we will look at incorporating intelligent optimization algorithms to see if we can lower the complexity of our method while maintaining its effectiveness. 

## Figures and Tables

**Figure 1 sensors-22-03667-f001:**
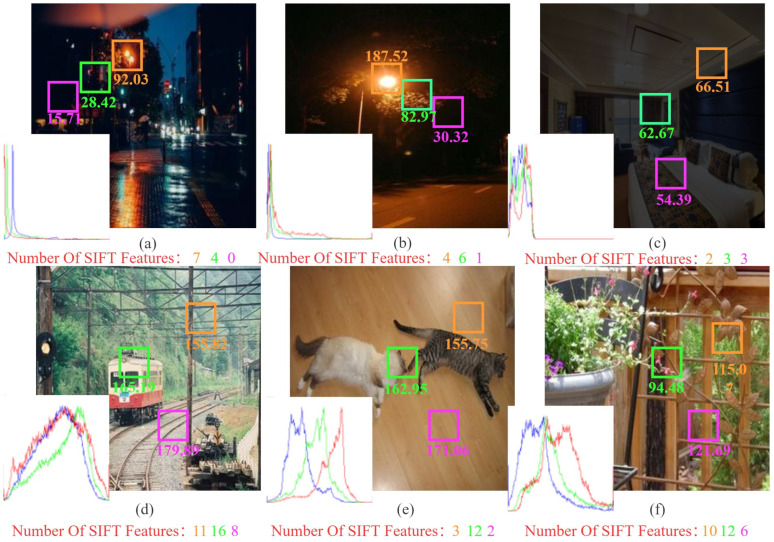
Examples of low-light images (top row) and bright images (bottom row) with global intensity histograms, their local illumination values and the number of sift features for each region. (**a**,**b**) Real low light images that exist in reality, (**c**) synthesized low light image, ((**d**–**f**) bright images).

**Figure 2 sensors-22-03667-f002:**
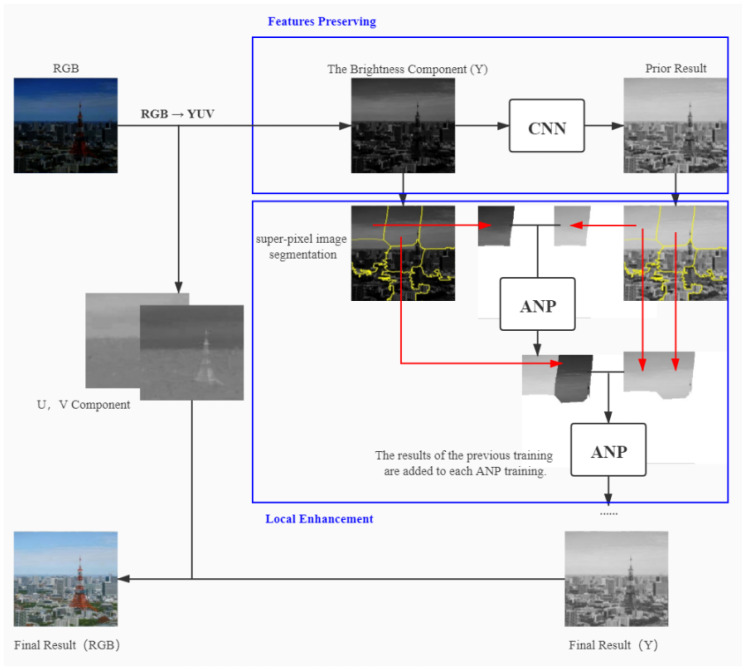
The flowchart of our method.

**Figure 3 sensors-22-03667-f003:**
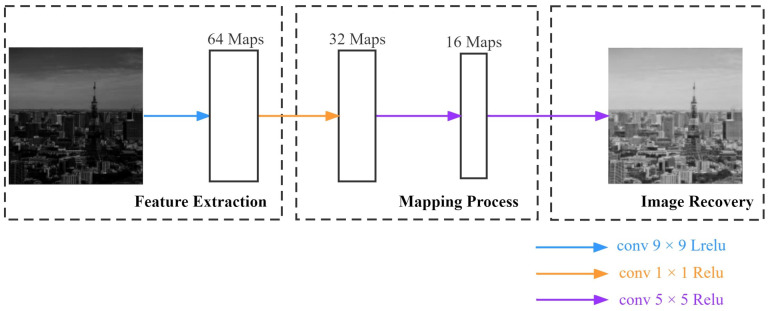
The architecture of CNN network.

**Figure 4 sensors-22-03667-f004:**
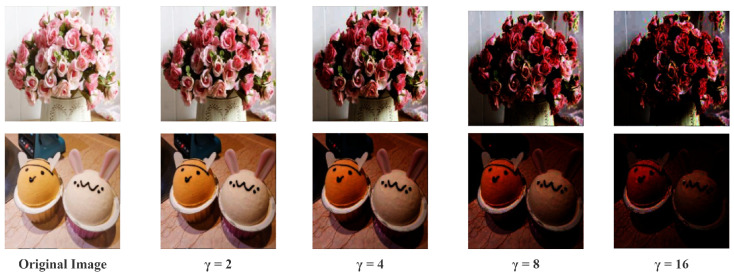
Low-light images are synthesized from bright images using different parameters γ.

**Figure 5 sensors-22-03667-f005:**
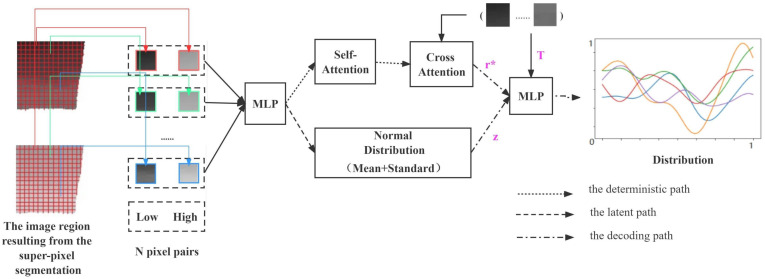
The structure of the ANP.

**Figure 6 sensors-22-03667-f006:**
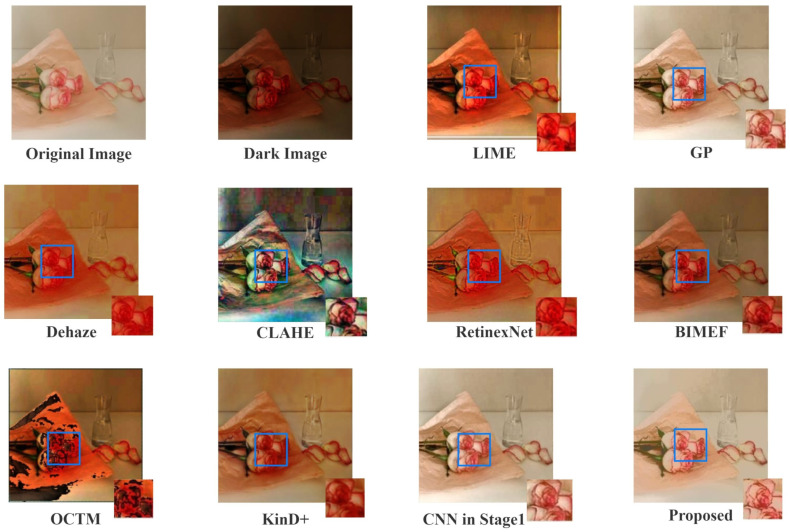
Example of low-light enhancement on a synthesized low-light image.

**Figure 7 sensors-22-03667-f007:**
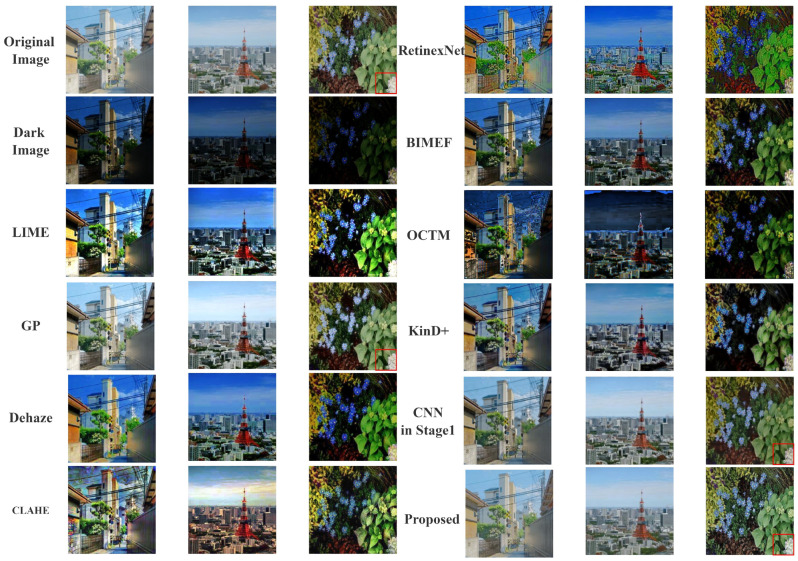
More examples of low-light enhancement on synthesized low-light images. The part represented by the red box has been enlarged and is shown in Figure 8.

**Figure 8 sensors-22-03667-f008:**
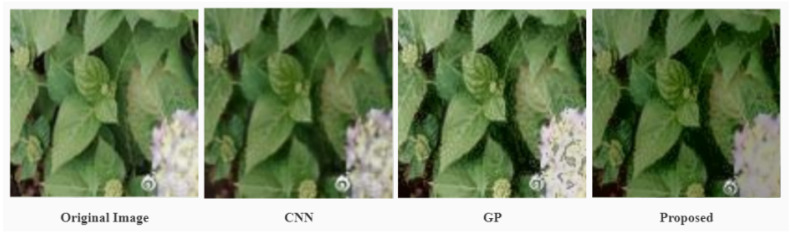
The comparison in details of GP, CNN, and our approach.

**Figure 9 sensors-22-03667-f009:**
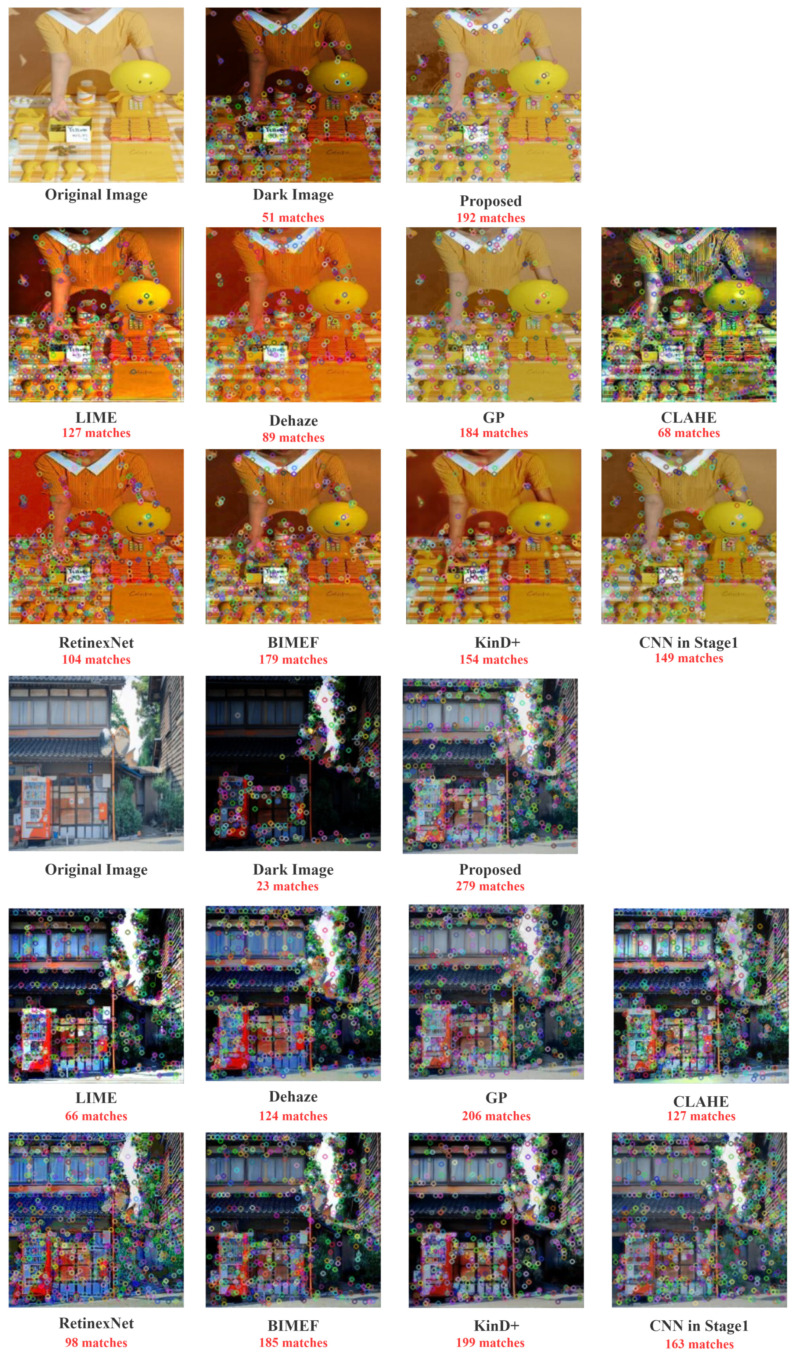
SIFT features matched in synthetic low-light images using different enhancement methods.

**Figure 10 sensors-22-03667-f010:**
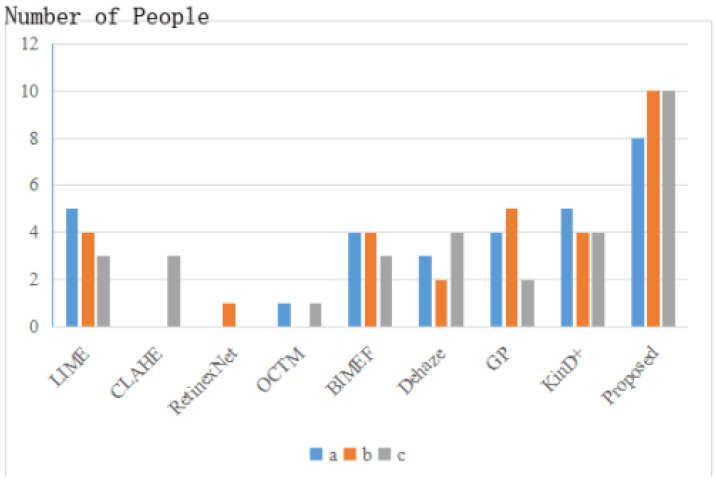
Subjective evaluation of enhanced results. a, b, c represent the enhancement results of three different real low light images shown in Figure 11.

**Figure 11 sensors-22-03667-f011:**
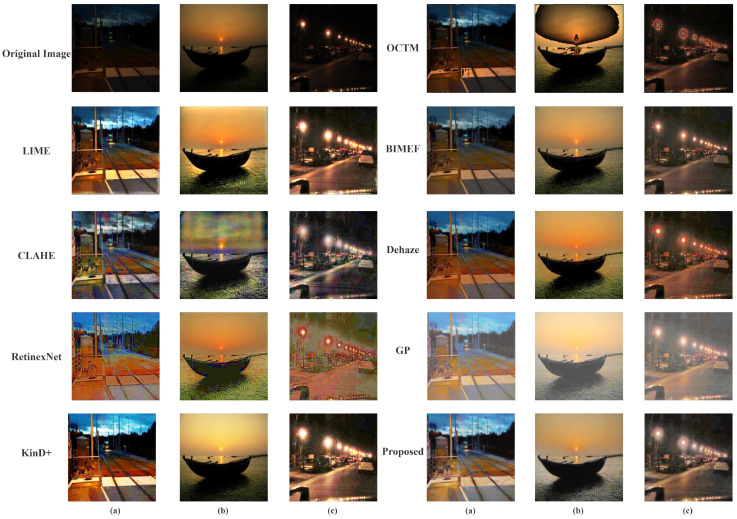
The enhancement results obtained for real images. They are the criteria for the experiments in Figure 10.

**Figure 12 sensors-22-03667-f012:**
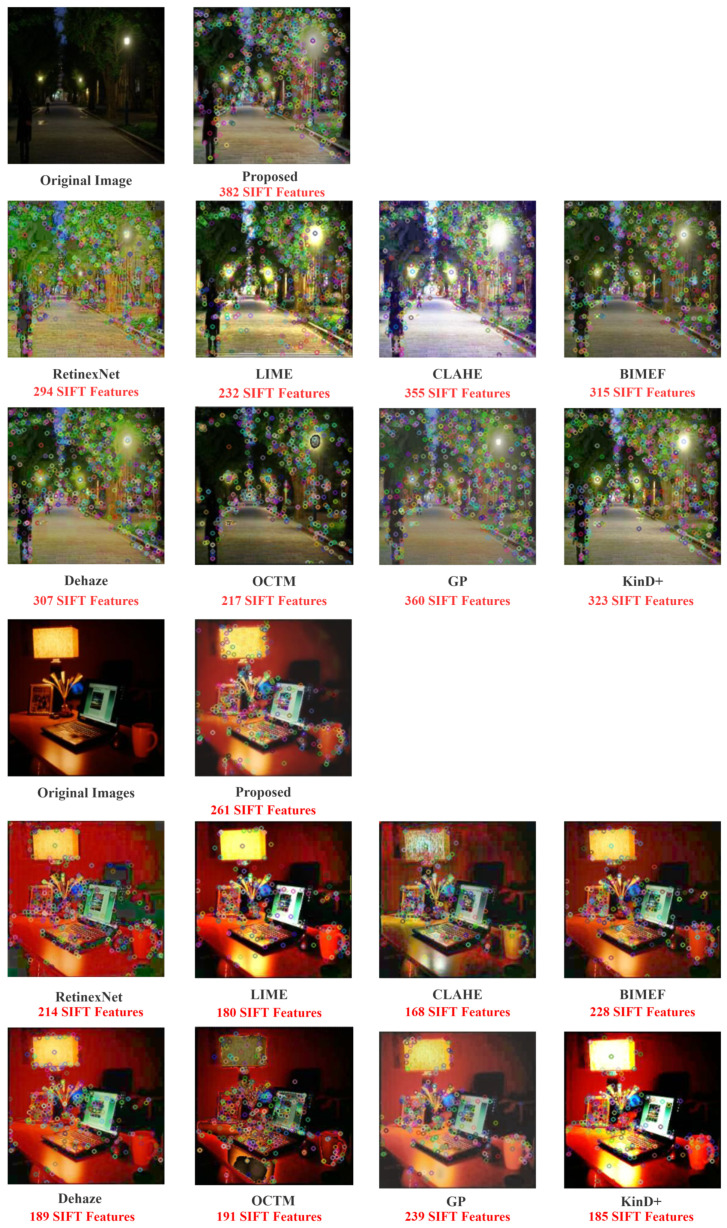
The enhanced image with their number of features.

**Figure 13 sensors-22-03667-f013:**
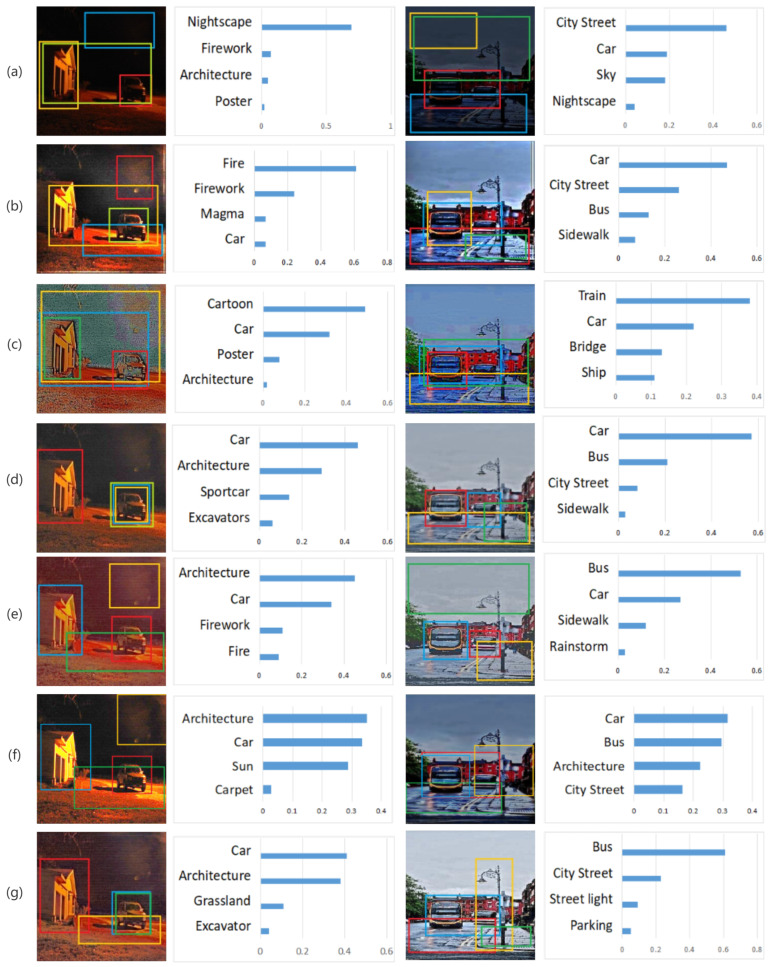
Top 4 object detection results of low-light image (**a**), enhanced by LIME (**b**), enhanced by RetinexNet (**c**), enhanced by CNN in stage 1 (**d**), enhanced by GP (**e**), KinD+ (**f**) and enhanced by our proposed method (**g**) using Yolov3 model. The detection rates from highest to lowest are: blue box, red box, yellow box and green box, respectively.

**Table 1 sensors-22-03667-t001:** Average PSNR and SSIM. The best data is marked in red.

Method	Dark Image	LIME	DeHaze	CLAHE	RetinexNet	KinD+
PSNR	12.523	16.345	17.369	19.014	16.223	23.016
SSIM	0.410	0.677	0.883	0.792	0.748	0.870
**Method**	**BIMEF**	**OCTM**	**GP**	**CNN in Stage 1**	**Remove Super Pixel Segmentation**	**Proposed**
PSNR	17.629	13.837	21.442	22.170	22.845	23.402
SSIM	0.866	0.629	0.891	0.902	0.898	0.916

**Table 2 sensors-22-03667-t002:** Average NIQE. The best data is marked in red.

Method	LIME	DeHaze	CLAHE	Retinex Net	GP	BIMEF	OCTM	KinD+	Proposed
NIQE	3.6680	3.3621	4.2952	3.5205	2.4387	2.5294	4.7017	2.5186	2.2490

## Data Availability

Not applicable.

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
