# Peer review of "Super-Pixel Guided Low-Light Images Enhancement with Features Restoration"

_sensors, 2022, doi:10.3390/s22103667_

Round 1

Reviewer 1 Report

Minor revision
This manuscript introduces a Low-Light images enhancement method, which uses a shallow Convolutional Neural Networks (CNNs) as the prior image processing to restore the necessary image feature information, and then incorporate super-pixel image segmentation technology to get image regions with similar colors and brightness, finally, the Attentive Neural Processes (ANPs) network is used to find its local enhancement function on each super-pixel to further restore features and details. In summary, the research is interesting and provides valuable results, but the current document has several weaknesses that must be strengthened in order to obtain a documentary result that is equal to the value of the publication.
General considerations:
(1)At the thematic level, the proposal provides a very interesting vision, as Low-Light images enhancement would be very helpful to make quality input data when the object detection to be used.
(2)The document contains a total of 37 employed references, of which 22 are publications produced in the last 5 years (59%), 2 in the last 5-10 years (5%), 13 than 10 years old (36%), implying a total percentage of 64 % recent references. To sum up, the references about related work is sufficient, but the author did not cite the latest algorithm for comparison experiments.
(3)Line 329 “The results of the average PSNR for the RGB channel and the SSIM for the gray channel for all test images. All values are obtained by randomly selecting 10 synthetic low-light images and enhancing them in the same running environment.” Using only 10 synthetic low-light images to prove the advantages of the paper may not be enough. where I think that the model needs more image data or more data groups to be set.
(4)The first paragraph introducing the research topic may present a much broad and comprehensive view of the problems related to your topic with citations to authority references (Tang, Y.; Zhu, M.; Chen, Z.; Wu, C.; Chen, B.; Li, C.; Li, L. Seismic Performance Evaluation of Recycled aggregate Concrete-filled Steel tubular Columns with field strain detected via a novel mark-free vision method. Structures, 2022, 37: 426-441. )
Title, Abstract and Keywords:
(5)The abstract is complete and well-structured and explains the contents of the document very well. Nonetheless, the part relating to the results could provide numerical indicators obtained in the research.
Chapter 1: Introduction
(6)On a general level, the study of the Low-Light images enhancement method is reasonable, and the explanation of the objectives of the work may be valid. However, the limitations of your work are not rigorously assumed and justified.
(7)Vision technology applications in various engineering fields, should also be introduced for a full glance of the scope of related area. For binocular vision, please refer to High-accuracy multi-camera reconstruction enhanced by adaptive point cloud correction algorithm; Real-time detection of surface deformation and strain in recycled aggregate concrete-filled steel tubular columns via four-ocular vision.
Chapter 3: The method
(8)Except in section 4.1, there appears to be no indication of the computational tools and software resources Including the use of image processing tools for other comparative experiments and the environment construction of confirmatory experiments.These issues could be presented in a more orderly and clear manner.
(9)Line 184 “Research manuscripts reporting large datasets that are deposited in a publicly avail-able database should specify where the data have been deposited and provide the relevant accession numbers. If the accession numbers have not yet been obtained at the time of submission, please state that they will be provided during review. They must be provided prior to publication.” These are not about papers. Don't write them.
Chapter 4: Experiments and results
(10)In the aspect of object detection test, only yolov3 is used as a tool for target detection of effect pictures. Therefore, it is suggested to add a group of high-version target detection tools to supplement the experiments, such as yolov5 or yolox to prove your results.
Chapter 5: Conclusions
(11)After all that has been read, this technique can be considered as a complement to images enhancement work, but it certainly does not seem to be a substitute for this work in its current state.
(12)It should mention the scope for further research as well as the implications/application of the study.

Reviewer 2 Report

Please check the highlighted sentences in the attached file. Line breaks are necessary after figure captions. US English and British English spelling words are mixed.

There are too many abbreviations in the manuscript. It makes readers difficult to follow. I recommend major revisions or rewriting.

Reviewer 3 Report

This paper proposes to combine CNN with super-pixel for low-light image enhancement. Both quantitative and qualitative experiments demonstrate the effectiveness of the proposed method. Some comments are as follows:
1. Does the CNN architecture affect the performance of the proposed method?
2. More recent methods are suggested to be compared in the experiments.
3. Figure 10 is confusing since many bars are missing. Please clarify this point. And the axis lacks explanations.
4. Related works about deep learning-based image restoration/quality enhancement and its quality metrics are recommended to be reviewed, including Learning disentangled feature representation for hybrid-distorted image restoration, ConvNet based single image deraining methods: A comparative analysis, Blind quality assessment for image superresolution using deep two-stream convolutional networks, Blind visual quality assessment for image super-resolution by convolutional neural network, etc.
5. Ablation study would be helpful to verify each component of the proposed framework.
6. The organization and presentation should be largely improved, such as the blurry figures, 255 255 pixels, etc. The title of Table 2 seems incorrect.

Round 2

Reviewer 2 Report

The revised version has been improved and easier to understand the aim of this study. I personally think that the role of super-pixel is not very clearly stated. I was able to get very similar results without using the super-pixel segmentation. However, I also understand that there can be many different methods can results very similar outcome. 

Please check the sentence in line 380-381 whether it was correctly stated. 

Reviewer 3 Report

The authors have well addressed my comments.

Author Response

Thank you for your positive comments on this paper.